# An Integrative Survey on Mental Health Conversational Agents to Bridge Computer Science and Medical Perspectives

**Young-Min Cho**[1]    **Sunny Rai**[1]    **Lyle Ungar**[1]
**João Sedoc**[2]    **Sharath Chandra Guntuku**[1]
[1]University of Pennsylvania    [2]New York University
{jch0,sunnyrai,ungar,sharathg}@seas.upenn.edu, jsedoc@stern.nyu.edu

## Abstract

Mental health conversational agents (a.k.a. chatbots) are widely studied for their potential to offer accessible support to those experiencing mental health challenges. Previous surveys on the topic primarily consider papers published in either computer science or medicine, leading to a divide in understanding and hindering the sharing of beneficial knowledge between both domains. To bridge this gap, we conduct a comprehensive literature review using the PRISMA framework, reviewing 534 papers published in both computer science and medicine. Our systematic review reveals 136 key papers on building mental health-related conversational agents with diverse characteristics of modeling and experimental design techniques. We find that computer science papers focus on LLM techniques and evaluating response quality using automated metrics with little attention to the application while medical papers use rule-based conversational agents and outcome metrics to measure the health outcomes of participants. Based on our findings on transparency, ethics, and cultural heterogeneity in this review, we provide a few recommendations to help bridge the disciplinary divide and enable the cross-disciplinary development of mental health conversational agents.

## 1 Introduction

The proliferation of conversational agents (CAs), also known as chatbots or dialog systems, has been spurred by advancements in Natural Language Processing (NLP) technologies. Their application spans diverse sectors, from education (Okonkwo and Ade-Ibijola, 2021; Durall and Kapros, 2020) to e-commerce (Shenoy et al., 2021), demonstrating their increasing ubiquity and potency.

The utility of CAs within the mental health domain has been gaining recognition. Over 30% of the world's population suffers from one or more mental health conditions; about 75% individuals in low and middle-income countries and about 50% individuals in high-income countries do not receive care and treatment (Kohn et al., 2004; Arias et al., 2022). The sensitive (and often stigmatized) nature of mental health discussions further exacerbates this problem, as many individuals find it difficult to disclose their struggles openly (Corrigan and Matthews, 2003).

Conversational agents like Woebot (Fitzpatrick et al., 2017) and Wysa (Inkster et al., 2018) were some of the first mobile applications to address this issue. They provide an accessible and considerably less intimidating platform for mental health support, thereby assisting a substantial number of individuals. Their effectiveness highlights the potential of mental health-focused CAs as one of the viable solutions to ease the mental health disclosure and treatment gap.

Despite the successful implementation of certain CAs in mental health, a significant disconnect persists between research in computer science (CS) and medicine. This disconnect is particularly evident when we consider the limited adoption of advanced NLP (e.g. large language models) models in the research published in medicine. While CS researchers have made substantial strides in NLP, there is a lack of focus on the human evaluation and direct impacts these developments have on patients. Furthermore, we observe that mental health CAs are drawing significant attention in medicine, yet remain underrepresented in health-applications-focused research in NLP. This imbalance calls for a more integrated approach in future studies to optimize the potential of these evolving technologies for mental health applications.

In this paper, we present a comprehensive analysis of academic research related to mental health conversational agents, conducted within the domains of CS and medicine[1]. Employing the Preferred Reporting Items for Systematic Reviews

---

[1]Our data and papers are available on our GitHub: https://github.com/JeffreyCh0/mental_chatbot_survey

and Meta-Analyses (PRISMA) framework (Moher et al., 2010), we systematically reviewed 136 pertinent papers to discern the trends and research directions in the domain of mental health conversational agents over the past five years. We find that there is a disparity in research focus and technology across communities, which is also shown in the differences in evaluation. Furthermore, we point out the issues that apply across domains, including transparency and language/cultural heterogeneity.

The primary objective of our study is to conduct a systematic and transparent review of mental health CA research papers across the domains of CS and medicine. This process aims not only to bridge the existing gap between these two broad disciplines but also to facilitate reciprocal learning and strengths sharing. In this paper, we aim to address the following key questions:

1. What are the prevailing focus and direction of research in each of these domains?

2. What key differences can be identified between the research approaches taken by each domain?

3. How can we augment and improve mental health CA research methods?

## 2 Prior Survey Papers

Mental health conversational agents are discussed in several non-CS survey papers, with an emphasis on their usability in psychiatry (Vaidyam et al., 2019; Montenegro et al., 2019; Laranjo et al., 2018), and users' acceptability (Koulouri et al., 2022; Gaffney et al., 2019). These survey papers focus on underpinning theory (Martinengo et al., 2022), standardized *psychological outcomes* for evaluation (Vaidyam et al., 2019; Gaffney et al., 2019) in addition to *accessibility* (Su et al., 2020), *safety* (Parmar et al., 2022) and *validity* (Pacheco-Lorenzo et al., 2021; Wilson and Marasoiu, 2022) of CAs.

Contrary to surveys for medical audiences, NLP studies mostly focus on the quality of the generated response from the standpoint of text generation. Valizadeh and Parde (2022) in their latest survey, reviewed 70 articles and investigated task-oriented healthcare dialogue systems from a technical perspective. The discussion focuses on the system architecture and design of CAs. The majority of healthcare CAs were found to have pipeline architecture despite the growing popularity of end-to-end architectures in the NLP domain. A similar technical review by Safi et al. (2020) also reports a high reliance on static dialogue systems in CAs developed for medical applications. Task-oriented dialogue systems usually deploy a guided conversation style which fits well with rule-based systems. However, Su et al. (2020); Abd-Alrazaq et al. (2021) pointed to the problem of robotic conversation style in mental health apps where users prefer an unconstrained conversation style and may even want to lead the conversation (Abd-Alrazaq et al., 2019). Huang (2022) further underlines the need for self-evolving CAs to keep up with evolving habits and topics during the course of app usage.

Surveys from the rest of CS cover HCI (de Souza et al., 2022) and the system design of CAs (Dev et al., 2022; Narynov et al., 2021a). de Souza et al. (2022) analyzed 6 mental health mobile applications from an HCI perspective and suggested 24 design considerations including *empathetic* conversation style, *probing*, and *session duration* for effective dialogue. Damij and Bhattacharya (2022) proposed three key dimensions namely *people* (citizen centric goals ), *process* (regulations and governance) and *AI technology* to consider when designing public care CAs.

These survey papers independently provide an in-depth understanding of advancements and challenges in the CS and medical domains. However, there is a lack of studies that can provide a joint

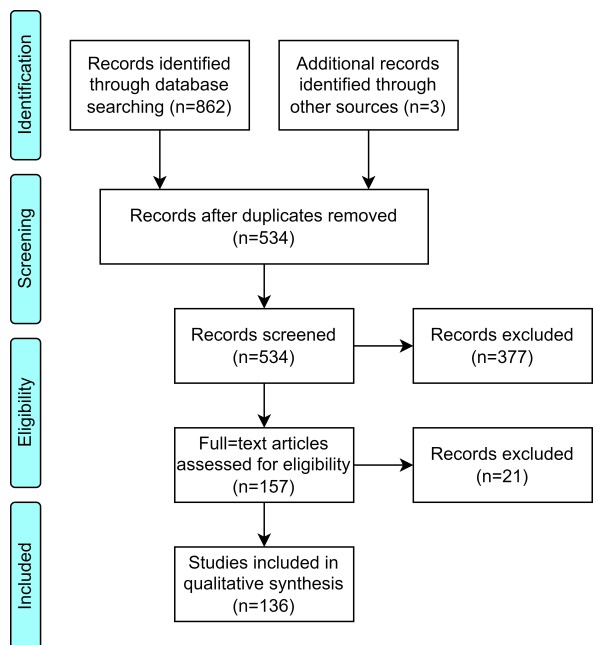

Figure 1: Pipeline of our PRISMA framework.

appraisal of developments to enable cross-learning across these domains. With this goal, we consider research papers from medicine (PubMed), NLP (the ACL Anthology), and the rest of CS (ACM, AAAI, IEEE) to examine the disparities in goals, methods, and evaluations of research related to mental health conversational agents.

## 3 Methods

### 3.1 Paper Databases

We source papers from eminent databases in the fields of NLP, the rest of CS, and medicine, as these are integral knowledge areas in the study of mental health CA. These databases include the ACL Anthology (referred to as ACL throughout this paper)[2], AAAI[3], IEEE[4], ACM[5], and PubMed[6]. ACL is recognized as a leading repository that highlights pioneering research in NLP. AAAI features cutting-edge studies in AI. IEEE, a leading community, embodies the forefront of engineering and technology research. ACM represents the latest trends in Human Computer Interaction (HCI) along with several other domains of CS. PubMed, the largest search engine for science and biomedical topics including psychology, psychiatry, and informatics among others provides extensive coverage of the medical spectrum.

Drawing on insights from prior literature reviews (Valizadeh and Parde, 2022; Montenegro et al., 2019; Laranjo et al., 2018) and discussion with experts from both the CS and medical domains, we opt for a combination of specific keywords. These search terms represent both our areas of focus: conversational agents ("conversational agent", "chatbot") and mental health ("mental health", "depression"). Furthermore, we limit our search criteria to the paper between 2017 to 2022 to cover the most recent articles. We also apply the "research article" filter on ACM search, and "Free Full Text or Full Text" for PubMed search. Moreover, we manually add 3 papers recommended by the domain experts (Fitzpatrick et al., 2017; Laranjo et al., 2018; Montenegro et al., 2019). This results in 534 papers.

### 3.2 Screening Process

For subsequent steps in the screening process, we adhere to a set of defined inclusion criteria. Specif-

---

[2] https://aclanthology.org/
[3] https://aaai.org/aaai-publications/
[4] https://ieeexplore.ieee.org/
[5] https://dl.acm.org/
[6] https://pubmed.ncbi.nlm.nih.gov/

| Screening Process | ACL | AAAI | IEEE | ACM | PubMed |
|---|---|---|---|---|---|
| Database Search | 68 | 30 | 52 | 280 | 104 |
| Title Screening | 26 | 16 | 39 | 137 | 84 |
| Abstract Screening | 9 | 4 | 31 | 45 | 68 |
| Full-Text Screening | **9** | **4** | **20** | **40** | **63** |
| Model / Experiment | **6** | **3** | **15** | **35** | **43** |

Table 1: Steps in the screening process and the number of papers retained in each database.

ically, we include a paper if it met the following conditions for a focused and relevant review of the literature that aligns with the objectives of our study:

- Primarily focused on CAs irrespective of modality, such as text, speech, or embodied.
- Related to mental health and well-being. These could be related to depression, PTSD, or other conditions defined in the DSM-IV (Bell, 1994) or other emotion-related intervention targets such as stress.
- Contribute towards directly improving mental health CAs. This could be proposing novel models or conducting user studies.

The initial step in our screening process is title screening, in which we examine all titles, retaining those that are related to either CA or mental health. Our approach is deliberately inclusive during this phase to maximize the recall. As a result, out of 534 papers, we keep 302 for the next step.

Following this, we proceed with abstract screening. In this stage, we evaluate whether each paper meets our inclusion criteria. To enhance the accuracy and efficiency of our decision-making process, we extract the ten most frequent words from the full text of each paper to serve as keywords. These keywords provide an additional layer of verification, assisting our decision-making process. Following this step, we are left with a selection of 157 papers.

The final step is full-text screening. When we verify if a paper meets the inclusion criteria, we extract key features (such as model techniques and evaluations) from the paper and summarize them in tables (see appendix). Simultaneously, we highlight and annotate the papers' PDF files to provide evidence supporting our claims about each feature

similar to the methodology used in Howcroft et al. (2020). This process is independently conducted by two co-authors on a subset of 25 papers, and the annotations agree with each other. Furthermore, the two co-authors also agree upon the definition of features, following which all the remaining papers receive one annotation.[7]

The final corpus contains 136 papers: 9 from ACL, 4 from AAAI, 20 from IEEE, 40 from ACM, and 63 from PubMed. We categorize these papers into four distinct groups: 102 model/experiment papers, 20 survey papers, and the remaining 14 papers are classified as 'other'. Model papers are articles whose primary focus is on the construction and explanation of a theoretical model, while experimental papers are research studies that conduct specific experiments on the models to answer pertinent research questions. We combine experiment and model papers together because experimental papers often involve testing on models, while model papers frequently incorporate evaluations through experiments. The 'other' papers include dataset papers, summary papers describing the proceedings of a workshop, perspectives/viewpoint papers, and design science research papers. In this paper, we focus on analyzing the experiment/model and survey papers, which have a more uniform set of features.

### 3.3 Feature Extraction

We extract a set of 24 features to have a detailed and complete overview of the recent trend. They include general features (*"paper type", "language", "mental health category", "background", "target group", "target demographic"*), techniques (*"chatbot name", "chatbot type", "model technique", "off the shelf", "outsourced model name", "training data"*), appearance (*"interface", "embodiment", "platform", "public access"*), and experiment (*"study design", "recruitment", "sample size", "duration", "automatic evaluation", "human evaluation", "statistical test", "ethics"*). Due to the limited space, we present a subset of the features in the main paper. Description of other features can be found in Appendix.[8]

### 4 Results

Under the category of model and experiment papers, there are 6 papers from ACL, 3 from AAAI,

---

[7]Annotated PDF files with evidence of each feature are available in our GitHub.

[8]Full feature table is available in the supplemental material.

| Language | CS | Med | All |
|---|---|---|---|
| English | 47 | 30 | 77 |
| Chinese | 1 | 5 | 6 |
| Korean | 4 | 1 | 5 |
| German | 1 | 1 | 2 |
| Italian | 1 | 1 | 2 |
| Portuguese | 0 | 2 | 2 |
| Other | 5 | 3 | 8 |

Table 2: Distribution of predominant language of the data and/or participants recruited in mental health CA papers. Other languages include Bangla, Danish, Dutch, Japanese, Kazakh, Norwegian, Spanish, and Swedish.

| Mental Health Category | CS | Med | All |
|---|---|---|---|
| Not Specified | 32 | 21 | 53 |
| Depression | 9 | 10 | 19 |
| Anxiety | 8 | 8 | 16 |
| Stress | 0 | 4 | 4 |
| Sexual Abuse | 3 | 0 | 3 |
| Social Isolation | 3 | 0 | 3 |
| Other | 14 | 11 | 25 |

Table 3: Distribution of mental health category in mental health CA papers. A paper could have multiple focused targets. Other categories include affective disorder, COVID-19, eating disorders, PTSD, substance use disorder, etc.

15 from IEEE, 35 from ACM, and 43 from PubMed. In this section, we briefly summarize the observations from the different features we extracted.

### 4.1 Language

We identify if there is a predominant language associated with either the data used for the models or if there is a certain language proficiency that was a part of the inclusion criteria for participants. Our findings, summarized in Table 2, reveal that English dominates these studies with over 71% of the papers utilizing data and/or participants proficient in English. Despite a few (17%) papers emerging from East Asia and Europe, we notice that studies in low-resource languages are relatively rare.

### 4.2 Mental Health Category

Most of the papers ( 43%) we reviewed do not deal with a specific mental health condition but work towards general mental health well-being (Saha et al., 2022a). The methods proposed in such papers are applicable to the symptoms associated with a broad range of mental health issues (e.g. emo-

| Target Demographic | CS | Med | All |
|---|---|---|---|
| General | 43 | 26 | 69 |
| Young People | 4 | 6 | 10 |
| Students | 5 | 3 | 8 |
| Women | 3 | 4 | 7 |
| Older adults | 4 | 1 | 5 |
| Other | 1 | 4 | 5 |

Table 4: Distribution of demographics focused by mental health CA papers. A paper could have multiple focused target demographic groups. Other includes black American, the military community, and employee.

| Model Technique | CS | Med | All |
|---|---|---|---|
| Retrieval-Based | 27 | 22 | 49 |
| Rule-Based | 23 | 19 | 42 |
| Generative | 10 | 0 | 10 |
| Not Specified | 3 | 3 | 6 |

Table 5: Distribution of model techniques used in mental health CA papers. A paper could use multiple modeling techniques. The Not Specified group includes papers without a model but employing surveys to ask people's opinions and suggestions towards mental health CA.

tional dysregulation). Some papers, on the other hand, are more tailored to address the characteristics of targeted mental health conditions. As shown in Table 3, depression and anxiety are two major mental health categories being dealt with, reflecting the prevalence of these conditions (Eagle et al., 2022). Other categories include stress management (Park et al., 2019; Gabrielli et al., 2021); sexual abuse, to help survivors of sexual abuse (Maeng and Lee, 2022; Park and Lee, 2021), and social isolation, mainly targeted toward older adults (Sidner et al., 2018; Razavi et al., 2022). Less-studied categories include affective disorders (Maharjan et al., 2022a,b), COVID-19-related mental health issues (Kim et al., 2022; Ludin et al., 2022), eating disorders (Beilharz et al., 2021), and PTSD (Han et al., 2021).

### 4.3 Target Demographic

Most of the papers (>65%) do not specify the target demographic of users for their CAs. The target demographic distribution is shown in Table 4. An advantage of the models proposed in these papers is that they could potentially offer support to a broad group of users irrespective of the underlying mental health condition. Papers without a target demographic and a target mental health category focus on proposing methods such as using generative language models for psychotherapy (Das et al., 2022a), or to address specific modules of the CAs such as leveraging reinforcement learning for response generation (Saha et al., 2022b). On the other hand, 31% papers focus on one specific user group such as young individuals, students, women, older adults, etc, to give advanced assistance. Young individuals, including adolescents and teenagers, received the maximum attention (Rahman et al., 2021). Several papers also

focus on the mental health care of women, for instance in prenatal and postpartum women (Green et al., 2019; Chung et al., 2021) and sexual abuse survivors (Maeng and Lee, 2022; Park and Lee, 2021). Papers targeting older adults are mainly designed for companionship and supporting isolated elders (Sidner et al., 2018; Razavi et al., 2022).

### 4.4 Model Technique

Development of Large Language Models such as GPT-series (Radford et al., 2019; Brown et al., 2020) greatly enhanced the performance of generative models, which in turn made a significant impact on the development of CAs (Das et al., 2022b; Nie et al., 2022). However, as shown in Table 5, LLMs are yet to be utilized in the development of mental health CAs (as of the papers reviewed in this study), especially in medicine. No paper from PubMed in our final list dealt with generative models, with the primary focus being rule-based and retrieval-based CAs.

Rule-based models operate on predefined rules and patterns such as if-then statements or decision trees to match user inputs with predefined responses. The execution of Rule-based CAs can be straightforward and inexpensive, but developing and maintaining a comprehensive set of rules can be challenging. Retrieval-based models rely on a predefined database of responses to generate replies. They use techniques like keyword matching (Daley et al., 2020), similarity measures (Collins et al., 2022), or information retrieval (Morris et al., 2018) to select the most appropriate response from the database based on the user's input. Generative model-based CAs are mostly developed using deep learning techniques such as recurrent neural networks (RNNs) or transformers, which learn from large amounts of text data and generate responses based on the learned patterns and struc-

| Outsourced Model | CS | Med | All |
|---|---|---|---|
| Google Dialogflow | 11 | 2 | 13 |
| Rasa | 5 | 5 | 10 |
| Alexa | 4 | 0 | 4 |
| DialoGPT | 3 | 0 | 3 |
| GPT | 3 | 0 | 3 |
| X2AI | 0 | 3 | 3 |
| Other | 17 | 6 | 23 |

Table 6: Distribution of outsourced models used for building models used in mental health CA papers. Other includes Manychat[9], Woebot (Fitzpatrick et al., 2017) and Eliza (Weizenbaum, 1966).

tures. While they can often generate more diverse and contextually relevant responses compared to rule-based or retrieval-based models, they could suffer from hallucination and inaccuracies (Azaria and Mitchell, 2023).

### 4.5 Outsourced Models

Building a CA model from scratch could be challenging for several reasons such as a lack of sufficient data, compute resources, or generalizability. Publicly available models and architectures have made building CAs accessible. Google Dialogflow (Google, 2021) and Rasa (Bocklisch et al., 2017) are the two most used outsourced platforms and frameworks. Alexa, DialoGPT (Zhang et al., 2019), GPT (2 and 3) (Radford et al., 2019; Brown et al., 2020) and X2AI (now called Cass) (Cass, 2023) are also frequently used for building CA models. A summary can be found in Table 6.

Google Dialogflow is a conversational AI platform developed by Google that enables developers to build and deploy chatbots and virtual assistants across various platforms. Rasa is an open-source conversational AI framework that empowers developers to create and deploy contextual chatbots and virtual assistants with advanced natural language understanding capabilities. Alexa is a voice-controlled virtual assistant developed by Amazon. It enables users to interact with a wide range of devices and services using voice commands, offering capabilities such as playing music, answering questions, and providing personalized recommendations. DialoGPT is a large, pre-trained neural conversational response generation model that is trained on the GPT2 model with 147M conversation-like exchanges from Reddit. X2AI is

[9]https://manychat.com

the leading mental health AI assistant that supports over 30M individuals with easy access.

### 4.6 Evaluation

**Automatic:** Mental health CAs are evaluated with various methods and metrics. Multiple factors, including user activity (total sessions, total time, days used, total word count), user utterance (sentiment analysis, LIWC (Pennebaker et al., 2015)), CA response quality (BLEU (Papineni et al., 2002), ROUGE-L (Lin, 2004), lexical diversity, perplexity), and performance of CA's sub-modules (classification f1 score, negative log-likelihood) are measured and tested. We find that papers published in the CS domain focus more on technical evaluation, while the papers published in medicine are more interested in user data.

**Human outcomes:** Human evaluation using survey assessment is the most prevalent method to gauge mental health CAs' performance. Some survey instruments measure the pre- and post-study status of participants and evaluate the impact of the CA by comparing mental health (e.g. PHQ-9 (Kroenke et al., 2001), GAD-7 (Spitzer et al., 2006), BFI-10 (Rammstedt et al., 2013)) and mood scores (e.g. WHO-5 (Topp et al., 2015)), or collecting user feedback on CA models (usability, difficulty, appropriateness), or asking a group of individuals to annotate user logs or utterances to collect passive feedbacks (self-disclosure level, competence, motivational).

### 4.7 Ethical Considerations

Mental health CAs inevitably work with sensitive data, including demographics, Personal Identifiable Information (PII), and Personal Health Information (PHI). Thus, careful ethical consideration and a high standard of data privacy must be applied in the studies. Out of the 89 papers that include human evaluations, approximately 70% (62 papers) indicate that they either have been granted approval by Institutional Review Boards (IRB) or ethics review committees or specified that ethical approval is not a requirement based on local policy. On the other hand, there are 24 papers that do not mention seeking ethical approval or consequent considerations in the paper. Out of these 24 papers that lack a statement on ethical concerns, 21 papers are published in the field of CS.

# 5 Discussion

## 5.1 Disparity in Research Focus

Mental health Conversational Agents require expert knowledge from different domains. However, the papers we reviewed, treat this task quite differently, evidenced by the base rates of the number of papers matching our inclusion criteria. For instance, there are over 28,000 articles published in the ACL Anthology with the keywords "chatbot" or "conversational agent", which reveals the popularity of this topic in the NLP domain. However, there are only 9 papers related to both mental health and CA, which shows that the focus of NLP researchers is primarily concentrated on the technical development of CA models, less on its applications, including mental health. AAAI shares a similar trend as ACL. However, there are a lot of related papers to mental health CAs in IEEE and ACM, which show great interest from the engineering and HCI community. PubMed represents the latest trend of research in the medical domain, and it has the largest number of publications that fit our inclusion criteria. While CS papers mostly do not have a specific focus on the mental health category for which CAs are being built, papers published in the medical domain often tackle specific mental health categories.

## 5.2 Technology Gap

CS and medical domains are also different in the technical aspects of the CA model. In the CS domain (ACL, AAAI, IEEE, ACM), 41 (of 73 papers) developed CA models, while 14 (out of 63) from the medical domain (PubMed) developed models. Among these papers, 8 from the CS domain are based on generative methods, but no paper in PubMed uses this technology. The NLP community is actively exploring the role of generative LLMs (e.g. GPT-4) in designing CAs including mental healthcare-related CAs (Das et al., 2022a; Saha et al., 2022b; Yan and Nakashole, 2021). With the advent of more sophisticated LLMs, *fluency*, *repetitions* and, *ungrammatical formations* are no longer concerns for dialogue generation. However, stochastic text generation coupled with black box architecture prevents wider adoption of these models in the health sector (Vaidyam et al., 2019). Unlike task-oriented dialogues, mental health domain CAs predominantly involve unconstrained conversation style for *talk-therapy* that can benefit from the advancements in LLMs (Abd-Alrazaq et al., 2021).

PubMed papers rather focus on retrieval-based and rule-based methods, which are, arguably, previous-generation CA models as far as the technical complexity is concerned. This could be due to a variety of factors such as explainability, accuracy, and reliability which are crucial when dealing with patients.

## 5.3 Response Quality vs Health Outcome

The difference in evaluation also reveals the varying focus across CS and medicine domains. From the CS domains, 30 (of 59 papers) applied automatic evaluation, which checks both model's performance (e.g. BLEU, ROUGE-L, perplexity) and participant's CA usage (total sessions, word count, interaction time). In contrast, only 13 out of 43 papers from PubMed used automatic evaluation, and none of them investigated the models' performance.

The difference is also spotted in human evaluation. 40 (of 43 papers) from PubMed consist of human outcome evaluation, and they cover a wide range of questionnaires to determine participants' status (e.g. PHQ-9, GAD-7, WHO-5). The focus is on users' psychological well-being and evaluating the chatbot's suitability in the clinical setup (Martinengo et al., 2022). Although these papers do not test the CA model's performance through automatic evaluation, they asked for participants' ratings to oversee their model's quality (e.g. helpfulness, System Usability Scale (Brooke et al., 1996), WAI-SR (Munder et al., 2010)).

All 6 ACL papers that satisfied our search criteria, solely focus on dialogue quality (e.g. *fluency*, *friendliness* etc.) with no discussion on CA's effect on users' well-being through clinical measures such as PHQ-9. CAs that aim to be the first point of contact for users seeking mental health support, should have clinically validated mechanisms to monitor the well-being of their users (Pacheco-Lorenzo et al., 2021; Wilson and Marasoiu, 2022). Moreover, the mental health CAs we review are designed without any underlying theory for psychotherapy or behavior change that puts the utility of CAs providing *emotional support* to those suffering from mental health challenges in doubt.

## 5.4 Transparency

None of the ACL papers that we reviewed released their model or API. Additionally, a *baseline* or comparison with the existing state-of-the-art model is often missing in the papers. There is no standard-

ized outcome reporting procedure in both medicine and CS domains (Vaidyam et al., 2019). For instance, Valizadeh and Parde (2022) raised concerns about the replicability of evaluation results and transparency for healthcare CAs. We acknowledge the restrictions posed to making the models public due to the sensitive nature of the data. However, providing APIs could be a possible alternative to enable comparison for future studies. To gauge the true advantage of mental health CAs in a clinical setup, randomized control trials are an important consideration that is not observed in NLP papers. Further, standardized benchmark datasets for evaluating mental health CAs could be useful in increasing transparency.

## 5.5 Language and Cultural Heterogeneity

Over 75% of the research papers in our review cater to English-speaking participants struggling with depression and anxiety. Chinese and Korean are the two languages with the highest number of research papers following English, even though Chinese is the most populous language in the world. Future works could consider tapping into a diverse set of languages that also have a lot of data available - for instance, Hindi, Arabic, French, Russian, and Japanese, which are among the top 10 most spoken languages in the world. The growing prowess of multilingual LLMs could be an incredible opportunity to transfer universally applicable development in mental health CAs to low-resource languages while being mindful of the racial and cultural heterogeneity which several multilingual models might miss due to being trained on largely English data (Bang et al., 2023).

## 6 Conclusion

In this paper, we used the PRISMA framework to systematically review the recent studies about mental health CA across both CS and medical domains. From the well-represented databases in both domains, we begin with 865 papers based on a keyword search to identify mental health-related conversational agent papers and use title, abstract, and full-text screening to retain 136 papers that fit our inclusion criteria. Furthermore, we extract a wide range of features from model and experiment papers, summarizing attributes in the fields of general features, techniques, appearance, and experiment. Based on this information, we find that there is a gap between CS and medicine in mental health CA

studies. They vary in research focus, technology, and evaluation purposes. We also identify common issues that lie between domains, including transparency and language/cultural heterogeneity.

## Potential Recommendations

We systematically study the difference between domains and show that learning from each other is highly beneficial. Since interdisciplinary works consist of a small portion of our final list (20 over 102 based on author affiliations on papers; 7 from ACM, 2 from IEEE, and 11 from PubMed), we suggest more collaborations to help bridge the gap between the two communities. For instance, NLP (and broadly CS) papers on mental health CAs would benefit from adding pre-post analysis on human feedback and considering ethical challenges by requesting a review of an ethics committee. Further, studies in medicine could benefit by tapping into the latest developments in generative methods in addition to the commonly used rule-based methods. In terms of evaluation, both the quality of response by the CAs (in terms of automatic metrics such as BLEU, ROUGE-L, perplexity, and measures of dialogue quality) as well as the effect of CA on users' mental states (in terms of mental health-specific survey inventories) could be used to assess the performance of mental health CAs. Moreover, increasing the language coverage to include non-English data/participants and adding cultural heterogeneity while providing APIs to compare against current mental health CAs would help in addressing the challenge of mental health care support with a cross-disciplinary effort.

## Limitations

This survey paper has several limitations. Our search criteria are between January 2017 to December 2022, which likely did not reflect the development of advanced CA and large language models like ChatGPT and GPT4 (Sanderson, 2023). We couldn't include more recent publications to meet the EMNLP submission date. Nonetheless, we have included relevant comments across the different sections on the applicability of more sophisticated models.

Further, search engines (e.g. Google Scholar) are not deterministic. Our search keywords, filters, and chosen databases do not guarantee the exact same search results. However, we have tested multiple times on database searching and they returned

consistent results. We have downloaded PDFs of all the papers and have saved the annotated them to reflect the different steps used in this review paper. These annotations will be made public.

For some databases, the number of papers in the final list may be (surprisingly!) small to represent the general research trends in the respective domains. However, it also indicates the lack of focus on mental health CA from these domains, which also proposes further attention is required in the field.

## Ethics Statement

Mental Health CAs, despite their accessibility, potential ability, and anonymity, cannot replace human therapists in providing mental health care. There are a lot of ongoing discussions about the range of availability of mental health CAs, and many raise several challenges and suspicions about automated conversations. Rule-based and retrieval-based models can be controlled for content generation, but cannot answer out-of-domain questions. Generative models are still a developing field, their non-deterministic nature raises concerns about the safety and reliability of the content. Thus at the current stage, CA could play a great supporting complementary role in mental healthcare to identify individuals who potentially need more immediate care in an already burdened healthcare system.

Since the patient's personal information and medical status are extremely sensitive, we highly encourage researchers and developers to pay extra attention to data security and ethics Arias et al. (2022). The development, validation, and deployment of mental health CAs should involve multiple diverse stakeholders to determine how, when, and which data is being used to train and infer participants' mental health. This effort requires a multidisciplinary effort to address the complex challenges of mental health care (Chancellor et al., 2019).

## Acknowledgements

We would like to thank the reviewers for their fruitful discussion with us. This work was partly supported by grant NIMHD: R01MD018340 from the National Institutes of Health and Penn Global Research Engagement Fund. The funders had no role in the design and conduct of the study; collection, management, analysis, and interpretation of the data; preparation, review, or approval of the manuscript; and decision to submit the manuscript for publication.

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

## A  Venues of Selected Papers

In this paper, we searched all venues indexed under 5 databases to cover most of the venues that are interested in mental health conversational agents. In Table 7, we show the distribution of venues under each database for the papers that are selected for the final list.

## B  Full Table Explanation

We show our final list of model/experiment papers in Table 8, Table 9 and Table 10. Due to the limited size of the paper, some columns ("background") are removed and long values are truncated. The full table is available on our GitHub.

For an easier understanding of our full table, we briefly introduce each feature we extracted below.

- *Paper*: The citation of the selected paper.
- *Database*: The source of the paper.
- *Paper Type*: The type of the paper. We here only show model or experiment papers.
- *Language*: Target language used in this paper.
- *Mental Health Category*: Target mental health category in this paper.
- *Target Group*: Target group of this paper. Could be patients, caregivers, or clinicians.
- *Target Demographic*: Target demographic of this paper. If it is not specified or can be used by anyone, we mark it as General.
- *Chatbot Name*: The name of the chatbot model used in this paper.
- *Chatbot Type*: Type of the mental health CA. Could be QA, open domain, or task-oriented.
- *Model Technique*: Type of technique used to build the model. Could be rule-based, retrieval-based, or generative.
- *Off the Shelf*: Information about the usage of off-the-shelf models in the system. We limit Off-the-shelf models to pre-trained models or

| AAAI | | ACL | | ACM | | IEEE | | PubMed | |
|---|---|---|---|---|---|---|---|---|---|
| Venue | Count | Venue | Count | Venue | Count | Venue | Count | Venue | Count |
| HCOMP | 2 | EMNLP | 1 | CHI | 9 | ICIRCA | 2 | JMIR Form Res | 9 |
| AAAI | 1 | SIGDIAL | 1 | ACM-TiiS | 4 | ACII | 2 | J Med Internet Res | 7 |
| | | BioNLP | 1 | IVA | 4 | IColCT | 1 | Front Digit Health | 4 |
| | | NAACL | 1 | ACM-HCI | 3 | UCET | 1 | JMIR Mhealth Uhealth | 3 |
| | | NLP4PI | 1 | UbiComp-ISWC | 2 | ICCCI | 1 | JMIR Res Protoc | 3 |
| | | LREC | 1 | CUI | 2 | ICHCI | 1 | Digit Health | 2 |
| | | | | PervasiveHealth | 2 | ICACCS | 1 | JMIR Ment Health | 2 |
| | | | | CHIItaly | 1 | ISCC | 1 | JMIR Hum Factors | 2 |
| | | | | ACSW | 1 | IEEE Trans. Emerg. | 1 | Internet Interv | 2 |
| | | | | H3 | 1 | SIEDS | 1 | Curr Psychol | 1 |
| | | | | Asian CHI | 1 | IEEE Pervasive Comput. | 1 | Comput Math Methods Med | 1 |
| | | | | DIS | 1 | ICCAS | 1 | Inf Process Manag | 1 |
| | | | | CHIuXiD | 1 | INCET | 1 | Front Psychol | 1 |
| | | | | ACM-HEALTH | 1 | | | Trials | 1 |
| | | | | IASA | 1 | | | Front Psychiatry | 1 |
| | | | | ECCE | 1 | | | Drug Alcohol Depend | 1 |
| | | | | | | | | Sensors (Basel) | 1 |
| | | | | | | | | JMIR Med Inform | 1 |

Table 7: Venues in each database that have at least one paper in our final list and the corresponding number of model/experiment papers.

applications. Could be yes (directly used), used as a part (off-the-shelf model is a part of the pipeline), or finetuned.

- *Outsourced Model Name*: The name of the off-the-shelf model, if any.
- *Training Data*: The name or source of the training data, if any.
- *Interface*: Type of input the model takes. Could be text, voice, visual, or button.
- *Embodiment*: Embodiment of the model. Could be physical or visual.
- *Platform*: The platform the model run on. Could be Web, Mobile, PC, or other devices.
- *Public Access*: If the availability of the model is disclosed in the paper. Could be fully open (parameter level) or API (able to use).
- *Study Design*: Type of user study performed in the paper. Could be RCT (Randomized Controlled Trial), user study (ask participants to use and evaluate), or comparative analysis (divide users with different conditions and compare the results).
- *Recruitment*: How participants are recruited.
- *Sample Size*: Size of the participants.
- *Duration*: Duration of the user study.
- *Automatic Evaluation*: List of automatic evaluation metrics used in this paper.
- *Human Evaluation*: List of parameters/metrics derived from Human Evaluation used in this paper.
- *Statistical Test*: List of statistical tests used for measuring significance in this paper.
- *Ethics*: Whether the paper mentioned ethical consideration. Could be IRB (Institutional Review Board), or yes (ethical consideration is mentioned in the paper).

Table 8: All method/experiment papers in the final list of this survey. This table only shows general and appearance features.

| Paper | Database | Paper Type | Language | Mental Health Category | Target Group | Target Demographic | Interface | Embodiment | Platform | Public Access |
|---|---|---|---|---|---|---|---|---|---|---|
| Abbas et al. (2020) | AAAI | Experiment | English | General | Clinicians | General | Text | | Web | / |
| Sun et al. (2022) | AAAI | Experiment | English | General | Clinicians | General | Text | | Web | / |
| Garg et al. (2020) | AAAI | Model | English | Depression | Patients | General | Text | | | / |
| Ishii et al. (2021) | ACL | Experiment | English | Isolation | Patients | General | Voice, Visual | Physical | Web | / |
| Demasi et al. (2020) | ACL | Model | English | Suicide | Clinicians | General | Text | | / | / |
| Das et al. (2022a) | ACL | Model | English | General | Patients | General | Text | | / | / |
| Saha et al. (2022b) | ACL | Model | English | General | Patients | General | Text | | / | / |
| Yan and Nakashole (2021) | ACL | Model | English | Well-being | Patients | General | Voice | Virtual | Web | / |
| van Waterschoot et al. (2020) | ACL | Model | Dutch | General | Patients | General | Voice | | Mobile, PC | / |
| Cox and Ooi (2022) | ACM | Experiment | English | General | Patients | General | Text | | PC | / |
| Fadhil et al. (2018) | ACM | Experiment | Italian | General | Patients | General | Voice, Visual | Virtual | PC | Fully Open |
| Jaiswal et al. (2019) | ACM | Experiment | English | Depression, Anxiety, Personality | Patients | General | Voice | Physical | Mobile, Smart Speaker | API |
| Maharjan et al. (2021) | ACM | Experiment | English | Depression, Anxiety | Patients | General | Text, Voice | /, Physical | Mobile, Smart Speaker | API |
| Eagle et al. (2022) | ACM | Experiment | Norwegian | General | Young People | General | Text | | Mobile, PC | API |
| Bae Brandtzaeg et al. (2021) | ACM | Experiment | Danish | Affective Disorder, Depression, Bipolar Disorder | Patients | General | Voice | Physical | Smart Speaker | API |
| Maharjan et al. (2022a) | ACM | Experiment | English | Depression, Anxiety | Patients | General | Text, Voice | /, Physical | Mobile, Smart Speaker | / |
| Quiroz et al. (2020) | ACM | Experiment | English | General | Patients | General | Text | | Mobile | API |
| Kawasaki et al. (2020) | ACM | Experiment | English | General | Patients | General | Voice | Physical | Mobile, Smart Speaker | / |
| Shin and Huh-Yoo (2020) | ACM | Experiment | English | Covid-19 | Patients | Black American | Voice | Physical | Mobile, PC | API |
| Kim et al. (2022) | ACM | Experiment | English | Depression, Anxiety | Patients | High School Students | Text | | Mobile, Other Devices | / |
| De Nieva et al. (2020) | ACM | Experiment | English | General | Patients | University Students | / | | / | API |
| Lee et al. (2020a) | ACM | Experiment | English | General | Patients | General | Text | Virtual | Mobile | / |
| Sweeney et al. (2021) | ACM | Experiment | English | General | Patients | General | Text | | Web, Mobile | / |
| Boyd et al. (2022) | ACM | Experiment | English | General | Patients | General | Text | | Mobile | API |
| Schroeder et al. (2018) | ACM | Model | English | Dialectical Behavior Therapy | Patients | General | Text | Virtual | Web, Mobile | / |
| Han et al. (2021) | ACM | Model | English | PTSD | Patients | General | Text | | Web, Mobile | / |
| Valtolina and Hu (2021) | ACM | Model | English | Loneliness | Patients | Elders | Text | | Mobile, PC | / |
| Sidner et al. (2018) | ACM | Model | English | Isolation | Patients | Older Adults | Text | Physical | Va, Robot | / |
| Luerssen and Hawke (2018) | ACM | Model | English | General | Patients | General | Text, Voice | Virtual | Mobile | API |
| Ryu et al. (2020) | ACM | Model | Korean | Depression, Anxiety | Patients | Older Adults | Text, Voice | | Mobile | / |
| Razavi et al. (2022) | ACM | Model | English | Isolation, Social Anxiety | Patients | Older Adults | Text | Virtual | Web | / |
| Lee et al. (2019) | ACM | Model | English | General | Patients, Caregivers | General | Text | | Mobile, PC | / |
| Holt-Quick and Warren (2021) | ACM | Model | English | General | Patients | General | Text | | Robot | / |
| Rastogi et al. (2018) | ACM | Model | English | Depression | Patients | General | Voice, Visual | Physical | Web | / |
| Ali et al. (2020) | ACM | Model | English | Autism Spectrum Disorder | Patients | Teenagers | Voice, Visual | Virtual | Mobile | / |
| Lee et al. (2020b) | ACM | Model | English | General | Patients | General | Text | | Mobile, PC | / |
| Sia et al. (2021) | ACM | Model | English | General | Patients | High School Students | Text | | Web | / |
| Park and Lee (2021) | ACM | Model | Korean | Sexual Assault | Patients | Women | Text | | Smart Speaker | / |
| Wang et al. (2020a) | ACM | Model | English | Public Speaking Anxiety | Patients | General | Voice | Physical | Mobile, PC | / |
| Park et al. (2021) | ACM | Model | Korean | Sharing Trauma | Patients | General | Text | | Mobile, Smart Speaker | / |
| Nie et al. (2022) | ACM | Model | English | General | Patients | General | Voice | /, Physical | Web, Mobile | / |
| Wang et al. (2021) | ACM | Model | Chinese | General | Patients | General | Text | | Mobile, PC | Fully Open |
| Rahman et al. (2021) | ACM | Model | Bangla | Sexual, Reproductive Health Problems | Patients | Adolescents | Text | | Mobile | / |
| Maeng and Lee (2022) | ACM | Model | Korean | Image-Based Sexual Abuse | Patients | Young Women | Text | | Mobile, PC | / |
| Ghandeharioun et al. (2019a) | IEEE | Experiment | English | General | Patients | General | Text | | Mobile | / |
| Siddik et al. (2022) | IEEE | Model | English | General | Patients | General | Text | | Mobile, PC | / |
| van Cuylenburg and Ginige (2021) | IEEE | Model | English | General | Parients | General | Text | | Web | / |
| Goel et al. (2021) | IEEE | Model | English | Depression, Anxiety | Patients | General | Text | | Mobile | / |
| Wang et al. (2020b) | IEEE | Model | English | Perinatal Mental Healthcare | Patients | Perinatal Women | Text | | | / |
| Dhanasekar et al. (2021) | IEEE | Model | English | General | Patients | Students | Text | | Mobile | / |
| Bhangdia et al. (2021) | IEEE | Model | English | General | Patients | General | Voice | | Web | / |
| Deepa et al. (2022) | IEEE | Model | English | General | Patients | General | Text | | Mobile | / |
| Potts et al. (2021) | IEEE | Model | English | General | Patients | General | Text | | Mobile, Web | API |
| Denecke et al. (2020) | IEEE | Model | German | General | Patients | General | Text | | Mobile | / |
| Ghandeharioun et al. (2019b) | IEEE | Model | English | General | Patients | General | Text | | Mobile | / |
| Schwartz et al. (2022) | IEEE | Model | English | Anxiety | Patients | General | Text | | Mobile | / |
| Maharjan et al. (2022b) | IEEE | Model | English | Affective Disorder | Patients | General | Voice | Physical | Smart Speaker | / |
| Narynov et al. (2021b) | IEEE | Model | Kazakh | General | Patients | General | Text | | / | / |
| Crasto et al. (2021) | IEEE | Model | English | General | Patients | Students | Text | | Mobile | / |
| Chan et al. (2022) | PubMed | Experiment | English | Eating Disorders | Patients | Adult Women | Text | | Mobile | / |
| Zhu et al. (2022) | PubMed | Experiment | Chinese | General | Patients | General | Text, Voice | | Mobile | API |

Table 8: All method/experiment papers in the final list of this survey. This table only shows general and appearance features.

| Paper | Database | Paper Type | Language | Mental Health Category | Target Group | Target Demographic | Interface | Embodiment | Platform | Public Access |
|---|---|---|---|---|---|---|---|---|---|---|
| Jiang et al. (2022) | PubMed | Experiment | Chinese | General | Patients | Women | Text | Virtual | Mobile, PC | API |
| Bennion et al. (2020) | PubMed | Experiment | English | General | Patients | Older Adults | Text | / | Web | / |
| Suganuma et al. (2018) | PubMed | Experiment | Japanese | General | Patients | General | Button | / | Web | / |
| Goonesekera and Donkin (2022) | PubMed | Experiment | English | Anxiety | Patients | General | Text | / | Mobile, PC | / |
| Gaffney et al. (2020) | PubMed | Experiment | English | General | Patients | General | Text | / | Web | / |
| Mariamo et al. (2021) | PubMed | Experiment | English | General | Patients | Adolescents | / | / | / | / |
| Provoost et al. (2020) | PubMed | Experiment | English | Low mood, Depression | Patients | General | Text | Virtual | Mobile, Web | / |
| Greer et al. (2019) | PubMed | Experiment | English | After Cancer Treatment | Patients | Young Adults | Text | / | Mobile, PC | / |
| Klos et al. (2021) | PubMed | Experiment | Spanish | Depression, Anxiety | Patients | General | Text | / | Mobile, PC | / |
| Liu et al. (2022) | PubMed | Experiment | Chinese | Depression | Patients | University Students | Text, Voice | / | Mobile, PC | API |
| Linden et al. (2020) | PubMed | Experiment | English | Anxiety, Depression, PTSD | Patients | Military Community | Text | / | Mobile | / |
| Gupta et al. (2022) | PubMed | Experiment | English | General | Patients | General | Text | / | Mobile | / |
| Prochaska et al. (2021a) | PubMed | Experiment | English | Substance Use Disorder | Patients | General | Text | / | Mobile, PC | API |
| Prochaska et al. (2021b) | PubMed | Experiment | English | Substance Use Disorder | Patients | General | Text | / | Mobile, PC | API |
| Darcy et al. (2021) | PubMed | Experiment | English | Depression, Anxiety | Patients | General | Text | / | Mobile, PC | / |
| Green et al. (2020) | PubMed | Experiment | English | Depression | Patients | Pregnant Women, New Mothers | Text | / | Mobile | / |
| Sinha et al. (2022) | PubMed | Experiment | English | General | Patients | General | / | / | Mobile | API |
| Schick et al. (2022) | PubMed | Experiment | German | Mental Disorders | Patients | Adolescence, Young Adulthood | Text, Button | / | PC | / |
| Beatty et al. (2022) | PubMed | Experiment | English | General | Patients | General | Text | / | Mobile | / |
| Meheli et al. (2022) | PubMed | Experiment | English | General | Patients | General | Text | / | Mobile | / |
| Dosovitsky et al. (2020) | PubMed | Experiment | English | General | Patients | General | Text | / | / | / |
| Dosovitsky et al. (2021) | PubMed | Experiment | English | Depression | Patients | General | Text | / | Mobile, PC | / |
| Hungerbuehler et al. (2021) | PubMed | Experiment | Portuguese | General | Patients | Employee | Text | Nan | Mobile, PC | / |
| Daley et al. (2020) | PubMed | Experiment | Portuguese | Anxiety, Depression, Stress | Patients | General | Text | Nan | Internet-Enabled Device | API |
| Ly et al. (2017) | PubMed | Experiment | Swedish | General | Patients | General | Text | / | Mobile | / |
| Gabrielli et al. (2021) | PubMed | Experiment | Italian | Stress, Anxiety | Patients | University Students | Text | / | Mobile, PC | API |
| He et al. (2022) | PubMed | Experiment | Chinese | General | Patients | Young Adults | Text | / | Mobile | / |
| Park et al. (2022) | PubMed | Model | English | General | Patients | General | Button | / | / | / |
| Hassan et al. (2021) | PubMed | Model | English | General | Patients | General | Text | / | Web | / |
| Burger et al. (2022) | PubMed | Model | English | Depression | Patients | General | Text | / | Mobile | / |
| De Gennaro et al. (2020) | PubMed | Model | English | Social Exclusion | Patients | General | Text, Button | Virtual | Web | / |
| Grové (2021) | PubMed | Model | English | General | Patients | Young People | Text | / | / | / |
| Park et al. (2019) | PubMed | Model | English | Stress | Patients | General | Text | / | Web | / |
| Rathnayaka et al. (2022) | PubMed | Model | English | General | Patients | General | Text | / | Mobile | API |
| Ludin et al. (2022) | PubMed | Model | English | Pandamic-Related Worry, Anxiety | Patients | Young People | Text | / | Web | / |
| Fitzpatrick et al. (2017) | PubMed | Model | English | Depression, Anxiety | Clinicians | University Students | Text | / | Mobile, PC | API |
| Noble et al. (2022) | PubMed | Model | English | General | Patients | Health Care Worker | Text | / | Web | / |
| Mauriello et al. (2021) | PubMed | Model | English | Stress | Patients | General | Text | / | Mobile | / |
| Chung et al. (2021) | PubMed | Model | Korean | General | Patients, Caregivers | Perinatal Womens, Partners | Text | / | Mobile | / |
| Morris et al. (2018) | PubMed | Model | English | General | Patients | General | Text | / | Mobile | API |
| Beilharz et al. (2021) | PubMed | Model | Chinese | Body Image, Eating Disorders | Patients | General | Button | / | Web | / |

Table 9: All method/experiment papers in the final list of this survey. This table only shows technique features. Long values are truncated due to limited space.

| Paper | Chatbot Name | Chatbot Type | Model Technique | Off the Shelf | Outsourced Model Name | Training Data |
|---|---|---|---|---|---|---|
| Abbas et al. (2020) | Trainbot | Task Oriented | Rule-Based | | | / |
| Sun et al. (2022) | MemberBot | QA | Retrieval-Based | Used As a Part | Rasa | / |
| Garg et al. (2020) | Unnamed | Open Domain | Retrieval-Based | | | (New) 7cups Conversation Data |
| Ishii et al. (2021) | ERICA, Nora | Task Oriented | Rule-Based | Used As a Part | Nora | Depression Therapy Sessions, L... |
| Demasi et al. (2020) | Crisisbot | Task Oriented | Generative, Retrieval-Based | | | Realistic Hotline Training Con... |
| Das et al. (2022a) | GPT2, DIALOGPT | Open Domain | Generative | Finetuned | GPT-2, DIALOGPT | (New) Reddit, Transcripts Of A... |
| Saha et al. (2022b) | MIC Model | Open Domain | Generative | Finetuned, Used As a Part | DialoGPT | (New) MotiVAte |
| Yan and Nakashole (2021) | SocialBot, Chatbot | Open Domain | Retrieval-Based, Generative | Finetuned, Used As a Part | GPT | (New) Medline Data, MedDialog... |
| van Waterschoot et al. (2020) | BLISS | Open Domain | Rule-Based, Retrieval-Based | Used As a Part | Flipper | (New) Collected From Users |
| Cox and Ooi (2022) | Unnamed | Task Oriented | Rule-Based | | | / |
| Fadhil et al. (2018) | CoachAi | Task Oriented | Rule-Based | | | / |
| Jaiswal et al. (2019) | ARIA-VALUSPA Platform | Task Oriented | | | | / |
| Maharjan et al. (2021) | Sofia | Task Oriented | Retrieval-Based | Used As a Part | Google Dialogflow | / |
| Eagle et al. (2022) | Google Assistant, Amazon Alexa... | / | / | Yes | Google Assistant, Amazon Alexa... | / |
| Bae Brandtzeg et al. (2021) | Woebot, Ungbot | Task Oriented | Rule-Based | Yes | Woebot, Ungbot | / |
| Maharjan et al. (2022a) | Sofia | Open Domain | Retrieval-Based | Used As a Part | Google Dialogflow | / |
| Quiroz et al. (2020) | Alexa Skill | Task Oriented | Generative | Yes | Alexa Skill | / |
| Kawasaki et al. (2020) | Unnamed | Task Oriented | Retrieval-Based | Used As a Part | Manychat, Google Dialogflow | / |
| Shin and Huh-Yoo (2020) | Alexa Skills | Task Oriented | Generative | Yes | Alexa Skill | / |
| Kim et al. (2022) | / | / | / | | | / |
| De Nieva et al. (2020) | Woebot | Task Oriented | Rule-Based | Yes | Woebot | / |
| Lee et al. (2020a) | Unnamed | Open Domain | Retrieval-Based | Used As a Part | Google Dialogflow | / |
| Sweeney et al. (2021) | / | Task Oriented | Retrieval-Based | | | (New) Use Cases Of Professiona... |
| Boyd et al. (2022) | ChatPal | Task Oriented | Retrieval-Based | Used As a Part | Rasa | Dr. Marsha Linehan's DBT Skill... |
| Schroeder et al. (2018) | Pocket Skills | Task Oriented | Rule-Based | | | Content From PTSD Coach |
| Han et al. (2021) | PTSDialogue | Task Oriented | Rule-Based | | | / |
| Valtolina and Hu (2021) | Charlie | Task Oriented | Rule-Based | Used As a Part | Google Dialogflow | / |
| Sidner et al. (2018) | AlwaysOn | Task Oriented | Rule-Based | | | / |
| Luerssen and Hawke (2018) | Clevertar | Task Oriented | Rule-Based | | | / |
| Ryu et al. (2020) | Yeonheebot | Task Oriented | Rule-Based | | | / |
| Razavi et al. (2022) | LISSA | Task Oriented | Retrieval-Based | | | / |
| Lee et al. (2019) | Vincent | Task Oriented | Retrieval-Based | Used As a Part | Google Dialogflow | / |
| Holt-Quick and Warren (2021) | Unnamed | Task Oriented | Retrieval-Based | Used As a Part | Rasa | / |
| Rastogi et al. (2018) | Unnamed | Task Oriented | Retrieval-Based | | | / |
| Ali et al. (2020) | LISSA | Task Oriented | Retrieval-Based | | | / |
| Lee et al. (2020b) | Unnamed | Task Oriented | Retrieval-Based | Used As a Part | Manychat, Google Dialogflow | / |
| Sia et al. (2021) | Abot | Task Oriented | Retrieval-Based | Used As a Part | Google Dialogflow | / |
| Park and Lee (2021) | NamuBot | Task Oriented | Rule-Based | | | / |
| Wang et al. (2020a) | Unnamed | Task Oriented | Retrieval-Based | Yes | Alexa | / |
| Park et al. (2021) | DIARYBOT | Task Oriented | Rule-Based | | | / |
| Nie et al. (2022) | Unnamed | Open Domain | Generative | Finetuned, Used As a Part | GPT-3 | (New) User Responses |
| Wang et al. (2021) | CASS | Open Domain | Generative | Finetuned, Used As a Part | OpenNMT | (New) Post-Response Pairs From... |
| Rahman et al. (2021) | AdolescentBot | Task Oriented | Retrieval-Based | Used As a Part | Wit.Ai | (New) Knowledge Base By Data F... |
| Maeng and Lee (2022) | Unnamed | Task Oriented | Rule-Based, Retrieval-Based | Used As a Part, Finetuned | BERT | (New) Emotional Support, Infor... |
| Ghandeharioun et al. (2019a) | EMMA | Task Oriented | Rule-Based | | | / |
| Siddik et al. (2022) | Unnamed | Task Oriented | Retrieval-Based | Used As a Part | Google DialogFlow | Reddit Mental Health Dataset |
| van Cuylenburg and Ginige (2021) | Unnamed | Task Oriented | Retrieval-Based | | | Kaggle |
| Goel et al. (2021) | Unnamed | Open Domain | Generative | | | Facebook AI Empathetic Dialogu... |
| Wang et al. (2020b) | Unnamed | Task Oriented | Rule-Based | | | / |
| Dhanasekar et al. (2021) | Maxx | Task Oriented | Rule-Based | Used As a Part | Google DialogFlow | / |
| Bhangdia et al. (2021) | Unnamed | Task Oriented | Rule-Based | | | / |
| Deepa et al. (2022) | Unnamed | Task Oriented | Retrieval-Based | | | / |
| Potts et al. (2021) | ChatPal | Task Oriented | Rule-Based | Used As a Part | Rasa | / |

Table 9: All method/experiment papers in the final list of this survey. This table only shows technique features. Long values are truncated due to limited space.

| Paper | Chatbot Name | Chatbot Type | Model Technique | Off the Shelf | Outsourced Model Name | Training Data |
|---|---|---|---|---|---|---|
| Denecke et al. (2020) | SERMO | Task Oriented | Retrieval-Based | Used As a Part | OSCOVA | / |
| Ghandeharioun et al. (2019b) | Unnamed | Task Oriented | Rule-Based | / | / | / |
| Schwartz et al. (2022) | DARA | Task Oriented | Retrieval-Based | Used As a Part, Finetuned | MindTrails | / |
| Maharjan et al. (2022b) | Sofia | Task Oriented | Retrieval-Based | Used As a Part | Google Dialogflow | / |
| Narynov et al. (2021b) | Unnamed | Task Oriented | Retrieval-Based | Used As a Part | Rasa | (New) Marked Entities In The D... |
| Crasto et al. (2021) | Carebot | Open Domain | Generative | Used As a Part, Finetuned | DialoGPT | (New) Data Scraped From Counse... |
| Chan et al. (2022) | Unnamed | Task Oriented | Rule-Based | Used As a Part | X2AI | Body Positive Conversations |
| Zhu et al. (2022) | Xiaolv | / | / | / | / | / |
| Jiang et al. (2022) | Replika | / | / | / | / | / |
| Bennion et al. (2020) | MYLO, ELIZA | Task Oriented | Rule-Based, Retrieval-Based | / | / | / |
| Suganuma et al. (2018) | SABORI | Task Oriented | Rule-Based | / | / | / |
| Goonesekera and Donkin (2022) | Otis | Task Oriented | Rule-Based | Yes | Chatfuel | / |
| Gaffney et al. (2020) | MYLO | Task Oriented | Retrieval-Based | / | / | / |
| Mariano et al. (2021) | / | Task Oriented | / | / | / | / |
| Provoost et al. (2020) | Moodbuster Lite | Task Oriented | Rule-Based | / | / | / |
| Greer et al. (2019) | Vivibot | Task Oriented | Rule-Based | / | / | / |
| Klos et al. (2021) | Tess | Task Oriented | Retrieval-Based | / | / | / |
| Liu et al. (2022) | XiaoNan | Task Oriented | Retrieval-Based | Used As a Part | Rasa | / |
| Linden et al. (2020) | Here4U App - Military Version | Task Oriented | Retrieval-Based | Yes | IBM's Watson Assistant | / |
| Gupta et al. (2022) | Wysa | Task Oriented | Rule-Based | / | / | / |
| Prochaska et al. (2021a) | W-SUDs (Weobot For SUDs) | Task Oriented | Rule-Based | / | / | / |
| Prochaska et al. (2021b) | Woebot | Task Oriented | Rule-Based | / | / | / |
| Darcy et al. (2021) | Woebot | Task Oriented | Rule-Based | / | / | / |
| Green et al. (2020) | Healthy Mons | Task Oriented | Rule-Based | Yes | Tess(Zuri) | / |
| Sinha et al. (2022) | Wysa | Task Oriented | Retrieval-Based | / | / | / |
| Schick et al. (2022) | Microfost Bot | Task Oriented | Retrieval-Based | / | / | / |
| Beatty et al. (2022) | Wysa | Task Oriented | Retrieval-Based | / | / | / |
| Meheli et al. (2022) | Wysa | Task Oriented | Retrieval-Based | / | / | / |
| Dosovitsky et al. (2020) | Tess | Task Oriented | Retrieval-Based | Yes | X2AI | / |
| Dosovitsky et al. (2021) | Tess | Task Oriented | Retrieval-Based | Yes | X2AI | / |
| Hungerbuehler et al. (2021) | Viki | Task Oriented | Rule-Based | / | / | / |
| Daley et al. (2020) | Vitalk | Task Oriented | Rule-Based | / | / | / |
| Ly et al. (2017) | Shim | Task Oriented | Rule-Based | / | / | (New) Professionals In Psychol... |
| Gabrielli et al. (2021) | Atena | Task Oriented | Rule-Based | / | / | (New) Psychologists |
| He et al. (2022) | XiaoE | Task Oriented | Retrieval-Based | Used As a Part | Rasa | (New) Psychologist Panel, Clin... |
| Park et al. (2022) | Unnamed | Task Oriented | Rule-Based | Used As a Part | Google DialogFlow | CDC's Mental Health Resourced |
| Hassan et al. (2021) | Unnamed | Task Oriented | Rule-Based | / | / | / |
| Burger et al. (2022) | Unnamed | Task Oriented | Rule-Based | Used As a Part | Rasa | / |
| De Gennaro et al. (2020) | Rose | / | Rule-Based | / | / | / |
| Grové (2021) | Ash | Task Oriented | Retrieval-Based | / | / | / |
| Park et al. (2019) | Bonobot | Task Oriented | Retrieval-Based | Used As a Part | ELIZA | / |
| Rathnayaka et al. (2022) | Bunji | Task Oriented | Retrieval-Based | Used As a Part | Rasa | / |
| Ludin et al. (2022) | Aroha | Task Oriented | Retrieval-Based | Used As a Part | Google DialogFlow | / |
| Fitzpatrick et al. (2017) | Woebot | Task Oriented | Rule-Based | / | / | / |
| Noble et al. (2022) | MIRA | Task Oriented | Retrieval-Based | Used As a Part | Rasa | (New) Study Team Members |
| Mauriello et al. (2021) | Popbots | Task Oriented | Retrieval-Based | / | / | (New) Workshop With Designers ... |
| Chung et al. (2021) | Dr. Joy | QA | Retrieval-Based | Yes | Kakao i | (New) Obstetric QA Knowledge D... |
| Morris et al. (2018) | Unnamed | Task Oriented | Retrieval-Based | / | / | (New) Koko Corpus |
| Beilharz et al. (2021) | KIT | Task Oriented | Rule-Based | / | / | (New) By The Authors |

Table 10: All method/experiment papers in the final list of this survey. This table only shows experiment features. Long values are truncated due to limited space.

| Paper | Study Design | Recruitment | Sample Size | Duration | Automatic Evaluation | Human Evaluation | Ethics | Statistical Test |
|---|---|---|---|---|---|---|---|---|
| Abbas et al. (2020) | Comparative Analysis | Prolific.Ac | 100, 100 | / | / | Enjoyment, Pressure, Helping S... | Yes | Independent Samples T-Test, Tw... |
| Sun et al. (2022) | User Study | MTurk Workers And Domain Exper... | 15, 11 | / | Number Of Messages, Length Of ... | Difficulty, Enjoyment | / | Mann–Whitney U Test, Linear Re... |
| Garg et al. (2020) | / | / | / | / | / | Appropriate, Diverse | / | / |
| Ishii et al. (2021) | Comparative Analysis | Recruited | 19 | / | Alignment | Overall Experience, Empathy, A... | / | One-Sided t-Test |
| Demasi et al. (2020) | User Study | Crowdworkers And Counslers | 30, 5 | / | Negative Log Likelihood, Entro... | Coherency, Consistency, Fluenc... | IRB | / |
| Das et al. (2022a) | User Study | Psychiatrist, Psychologist | 1,1 | / | Lexical Diversity, Average Cos... | Communication, Basic Psychothe... | / | Cohen's , Krippendorff's |
| Saha et al. (2022b) | User Study | Recruited | 3 | / | BLEU-1, Perplexity, ROUGE-L, E... | Fluency, Adaptability, Motivat... | Yes | Welch's t-Test |
| Yan and Nakashole (2021) | / | / | / | / | Accuracy, Negative Log Likehoo... | / | Yes | / |
| van Waterschoot et al. (2020) | / | / | / | / | / | / | Yes | / |
| Cox and Ooi (2022) | Comparative Analysis | Amazon Mechanical Turk | 187, 156 | / | Word Count | Likelihood To Disclose, Enjoym... | / | Tukey's HSD |
| Fadhil et al. (2018) | Comparative Analysis | Recruited | 58 | / | Interaction Time | Individual Self-Confidence And... | / | Mixed-Design ANOVA |
| Jaiswal et al. (2019) | Comparative Analysis | Reached Out | 55 | / | / | PHQ-9, GAD-7, BFI-10 | / | Two One Sided t-Test |
| Maharjan et al. (2021) | Comparative Analysis | Recruited From a Local Univers... | 59 | / | Completion Time, Correlation B... | SASSI Scores, WHO-5 | / | Fleiss' Kappa, Nonparametric M... |
| Eagle et al. (2022) | Comparative Analysis | Trained Researchers, Mental He... | 4, 2 | 2 Weeks | / | PHQ-8, GAD-7, Treatment, Empah... | / | Shapiro-Wilks Test, Levene's T... |
| Bae Brandtzeg et al. (2021) | User Study | Recruited In Universities | 16 | 4 Weeks | / | Appraisal Support, Emotional S... | Yes | / |
| Maharjan et al. (2022a) | User Study | National Patient Recruitment S... | 20 | 2 Weeks | / | User Experience Questionnaire,... | Yes | / |
| Quiroz et al. (2020) | User Study | Recruited | 10 | 2 Weeks | / | PHQ-9, GAD-7, System Usability... | Yes | / |
| Kawasaki et al. (2020) | Comparative Analysis | Social Media, Websites, Univer... | 30 | 3 Weeks | Word Counts, Use Of Positive/N... | Kessler Psychological Distress... | IRB | Mixed-Model ANOVA, Tukey HSD |
| Shin and Huh-Yoo (2020) | User Study | Users | 1 | / | / | (1) Reasons For Reviewers Usin... | IRB | / |
| Kim et al. (2022) | User Study | University's Health Clinic, Em... | 18 | / | / | Roles, Features, And Challenge... | IRB | / |
| De Nieva et al. (2020) | RCT | Senior High School Students | 25 | 2 Weeks | / | Psychological Distress Assessm... | / | Wilcoxon Signed Rank Test |
| Lee et al. (2020a) | Comparative Analysis | University Students | 47 | 4 Weeks | / | Self-Disclosure Level, Constru... | IRB | Mixed Model ANOVA |
| Sweeney et al. (2021) | User Study | Experts In Mental Health | 100 | / | / | Usage Of Chatbot, Benefits, Ch... | Yes | Spearman's Rank Correlation Co... |
| Boyd et al. (2022) | User Study | Action Mental Health And Ulste... | 10 | / | / | System Usability Scale, Chatbo... | Yes | Kruskal-Wallis Test, Pearson C... |
| Schroeder et al. (2018) | User Study | Resruited Via a DBT Listserve | 73 | 4 Weeks | Completion Time, Success Propo... | OASIS, PHQ-9, DBT WOCC | IRB | Linear Regression |
| Han et al. (2021) | / | / | / | / | / | / | Yes | / |
| Valtolina and Hu (2021) | User Study | Students' Relatives | 12 | 1 Week | Total Sessions, Total Time, Da... | Perceptions, Acceptance, Perce... | / | Non-Parametric Mann–Whitney Te... |
| Sidner et al. (2018) | Comparative Analysis | Craigslist's Posts, Fliers, Pr... | 44 | a Month | / | Sociodemographic Questionnaire... | / | / |
| Luerssen and Hawke (2018) | User Study | Google, Facebook, a Network Of... | 163 | 6 Weeks | Monetary, Dementia Information... | Kessler Psychological Distress... | / | Two-Tailed Paired t-Test |
| Ryu et al. (2020) | User Study | Clinic, Elderly Center, Welfar... | 24, 25 | 1 Day, 2 Weeks | Elaborateness, Sentiment Analy... | Center For Epidemiologic Studi... | / | Two-Tailed t-Test |
| Razavi et al. (2022) | RCT | Community Advertisement, Outpa... | 20 | 3-4 Weeks | / | / | / | Pearson r |
| Lee et al. (2019) | Comparative Analysis | Participant Database | 12, 67 | 3 Days, 2 Weeks | Error Rate, Total Word Count | Self-Compassion Scale, Irregul... | IRB | One-Tailed Independent t-Tests... |
| Holt-Quick and Warren (2021) | / | / | / | / | / | The Ability To Learn The Speci... | IRB | / |
| Rastogi et al. (2018) | / | / | / | / | / | / | / | / |
| Ali et al. (2020) | User Study | Through The Developmental/Beha... | 47, 9 | / | / | Perceptions, Re... | / | Non-Parametric Mann–Whitney U ... |
| Lee et al. (2020b) | Comparative Analysis | Social-Media Websites, Univers... | 47 | 3 Weeks | Word Count, Word Length | Usefulness, Perceptiveness, Re... | Yes | Mixed-Model ANOVA, Tukey HSD, ... |
| Sia et al. (2021) | User Study | Convenience Sampling, Email In... | 25 | 1 Week | Completion Rate | Categories And Levels, Trust,... | Yes | / |
| Park and Lee (2021) | User Study | Social Media, Personal Contact... | 19 | / | / | Performance, Humanity, Affect,... | IRB | / |
| Wang et al. (2020a) | User Study | Reached Out To Students In Pub... | 53 | / | / | Burdens Placed By Chatbot | IRB | / |
| Park et al. (2021) | Comparative Analysis | University Students | 30 | 4 Days | / | State Public Speaking Anxiety,... | IRB | Paired Sample t-Test, Mediatio... |
| Nie et al. (2022) | User Study | Volunteers | 7 | 1 Week | Dimension Classification Accur... | Schwartz Outcome Scale, Clinic... | IRB | One-Way ANOVA, Post-Hoc Tukey ... |
| Wang et al. (2021) | User Study | Recruited | 5 | / | BLEU | Overall Scoring, Willingness, ... | IRB | Independent Sample t-Test |
| Rahman et al. (2021) | User Study | Schools, Colleges, University | 256 | / | / | Grammar Correctness, Relevance... | Yes | / |
| Maeng and Lee (2022) | Comparative Analysis | Recruited | 25 | / | / | Effectiveness, Consistency, Pe... | IRB | One-Tailed Paired t-Tests |
| Ghandeharioun et al. (2019a) | RCT | Part Of The Bigger Project | 39 | 2 Weeks | Response Latency, Frequency Of... | 1) Accessibility, 2) Appropria... | IRB | Pearson Correlation Coefficien... |
| Siddik et al. (2022) | User Study | Recruited | 24 | / | Classification Accuracy | User Preference | IRB | / |
| van Cuylenburg and Ginige (2021) | / | / | / | / | Precision, Recall, F1-Score, S... | PHQ-9, GAD-7 | / | / |
| Goel et al. (2021) | / | / | / | / | BLEU | / | / | / |
| Wang et al. (2020b) | / | / | / | / | / | EPDS, WEMWBS | / | / |
| Dhanasekar et al. (2021) | Comparative Analysis | From a College | 40 | / | Accuracy, Precision, Recall, F... | Performance | Yes | / |
| Bhangdia et al. (2021) | / | / | / | / | Accuracy, Precision, Recall, F... | / | / | / |
| Deepa et al. (2022) | User Study | Users | 211 | / | User Tenure, Unique Days, Tota... | WHO-5 | Yes | / |
| Potts et al. (2021) | User Study | Nan | 21 | / | / | User Experience Questionnaire | Yes | / |
| Denecke et al. (2020) | / | / | / | / | / | / | / | / |
| Ghandeharioun et al. (2019b) | Comparative Analysis | Part Of The Bigger Project | 39 | 1 Week | Experience Sampling | Big Five Personality Traits, P... | IRB | Pearson Correlation Coefficien... |
| Schwartz et al. (2022) | User Study | Subject-Matter Exports | 12 | / | Chatbot Session Length And Cha... | PSSUQ, 10 Additional Quantitat... | IRB | / |
| Maharjan et al. (2022b) | Comparative Analysis | National Recruitment Site Http... | 22 | 4 Weeks | Sentiment Analysis | User Experience Questionnaire,... | Yes | Welch Two Sample t-Test |
| Narynov et al. (2021b) | / | / | / | / | Accuracy | / | / | / |
| Crasto et al. (2021) | Comparative Analysis | Recruited | 100 | / | / | PHQ-9, GAD-7 | / | / |
| Chan et al. (2021) | User Study | Social Media, Flyers, Referral... | 210 | 1 Week | / | Weight Concerns Scale, Stanfor... | IRB | / |
| Zhu et al. (2022) | User Study | WeChat Groups | 371 | / | / | Personalization, Voice Interac... | / | Partial Least Squares Structur... |

Table 10: All method/experiment papers in the final list of this survey. This table only shows experiment features. Long values are truncated due to limited space.

| Paper | Study Design | Recruitment | Sample Size | Duration | Automatic Evaluation | Human Evaluation | Ethics | Statistical Test |
|---|---|---|---|---|---|---|---|---|
| Jiang et al. (2022) | RCT | Advertised Over The Web, Poste... | / | 2 Weeks | Time | Related Social Media Posts | Yes | ANOVA, Independent t Tests Tha... |
| Bennion et al. (2020) | Comparative Analysis | Internet Research Company | 112 | 1 Month | / | Personal Problems, Helpfulness... | Yes | Two-Factor Mixed Model ANOVA |
| Suganuma et al. (2018) | Comparative Analysis | Facebook, Instagram, Twitter, ... | 191, 263 | 2 Weeks | Adherence | WHO-5-J, K19, BADS-AC, BADS-AR | Yes | Paired Samples t Tests And 1-W... |
| Goonesekera and Donkin (2022) | User Study | Email, Telephone | 29 | 2 Weeks | Frequency, Duration | SHAI-18, GAD-7, IUS-12, ONS4... | Yes | Power Analysis, Paired Samples... |
| Gaffney et al. (2020) | User Study | Flyers And Facebook Advertisem... | 15 | / | / | Helpfulness, Key Mechanisms Of... | Yes | Panel Logistic Regressions |
| Mariamo et al. (2021) | Comparative Analysis | Advertisements In Digital Medi... | 19 | 4 Weeks | Adherence | Perceived Emotionla Valence, L... | Yes | Point Estimates, General Linea... |
| Provoost et al. (2020) | RCT | Facebook, Usrvivorship Organiz... | 35, 35 | 4 Weeks | Time Spent On All Sessions | Short Motivation Feedback List. | Yes | Chi-Square Test, t-Test |
| Greer et al. (2019) | RCT | Presentations In University Co... | 51 | 8 Weeks | / | Engagement With The Chatbot, C... | Yes | Mann-Whitney U And Wilcoxon Te... |
| Klos et al. (2021) | RCT | Online Poster | 39, 34 | 16 Weeks | / | PHQ-9, GAD-7 | Yes | Independent t-Tests And Chi-Sq... |
| Liu et al. (2022) | RCT | Snowball Sampling | 83 | / | / | PHQ-9, GAD-7 (Spitzeret Al., 2... | Yes | Wilcoxon Signed-Rank Test, Pai... |
| Linden et al. (2020) | User Study | Internet Communities | 93 | 8 Weeks | / | Usability, Suggestions, Identi... | Yes | Paired Samples t-Tests And Chi... |
| Gupta et al. (2022) | User Study | Qualtrics, Stanford Listservs,... | / | 8 Weeks | / | NPRS, PROMIS-PI, PHQ-9, GAD-7... | Yes | Paired Samples t Tests And McN... |
| Prochaska et al. (2021a) | RCT | User, Social Media, Craigslist... | 180 | 8 Weeks | / | Change In Past-Month Substance... | IRB | Spearman Rank-Order Correlatio... |
| Prochaska et al. (2021b) | User Study | User | 101 | 8 Weeks | / | The Alcohol Use Disorders Iden... | IRB | Bayesian Linear Mixed-Effects ... |
| Darcy et al. (2021) | User Study | Hospital | 36070 | 5 Days | / | PHQ-2, Working Alliance Invent... | IRB | Kaplan-Meier Nonparametric Est... |
| Green et al. (2020) | User Study | US Tertiary Care Orthopedic Cl... | 10 | 1-2 Weeks | Intervention Use | Feasibility, Acceptability, De... | IRB | ANOVA, Repeated-Measures ANOVA... |
| Sinha et al. (2022) | User Study | University's Research Panel | 49 | 8 Weeks | App's Usage Log, Number Of Ses... | Experience, Balanced Inventory. | IRB | The Wilcoxon Signed Rank Test |
| Schick et al. (2022) | Comparative Analysis | New Users | 146 | / | / | Working Alliance Inventory-Sho... | Yes | Mann-Whitney U Test, Paired t ... |
| Beatty et al. (2022) | User Study | Users | 1205 | 3 Days | Textual Snippets From Users | PHQ-9, GAD-7 | Yes | / |
| Meheli et al. (2022) | User Study | Users | 2194 | / | Textual Snippets, Tool Usage, ... | / | Yes | Cronbach's Alpha, Spearman's R... |
| Dosovitsky et al. (2020) | User Study | Users | 354 | / | Total Messages Sent From/To Us... | PHD-9, Usefulness | Yes | Cohen's d, Standardized Coeffi... |
| Dosovitsky et al. (2021) | User Study | Facebook | 3895 | 6 Month | / | PHD-9, GAD-7, DASS-21, Insomni... | Yes | Independent t-Tests And X2-Tes... |
| Hungerbuehler et al. (2021) | User Study | Email, Intranet, Banners, Leaf... | 77 | / | / | PHD-9, GAD-7, DASS-21 | Yes | Shapiro Test, Paired-Samples t... |
| Daley et al. (2020) | User Study | User | 3629 | 90 Days | / | Flourishing Scale, The Satisfa... | IRB | G* Power, Analysis Of Covarian... |
| Ly et al. (2017) | RCT | Recruited From University | 14, 14 | 2 Weeks | / | Perceived Stress Scale, Genera... | IRB | Cronbach's, And Correlation ... |
| Gabrielli et al. (2021) | User Study | Social Media Outlets, Online P... | 71 | 4 Weeks | / | PHQ-9, Diagnostic AndStatistic... | Yes | / |
| He et al. (2022) | RCT | Amazon Mechanical Turk | 148 | 1 Week | / | Chatbot Emotional Disclosure, ... | / | Spearman's p |
| Park et al. (2022) | Comparative Analysis | / | 348 | / | / | / | / | Independent Samples t-Test, AN... |
| Hassan et al. (2021) | / | / | / | / | / | PHQ-9, Engagement In Self-Refl... | Yes | / |
| Burger et al. (2022) | Comparative Analysis | Prolific, a Crowd-Sourcing Pla... | 308 | Nan | / | Positive And Negative Affect S... | Yes | / |
| De Gennaro et al. (2020) | Comparative Analysis | Department Subject Pool | 64, 64 | / | / | Participants' Interests And Th... | Yes | / |
| Grové (2021) | User Study | Recruited | 40 | / | / | Perceived Stress Scale (PSS-10... | Yes | / |
| Park et al. (2019) | User Study | University Online Bulletin | 30 | / | / | PHQ-9 | IRB | / |
| Rathnayaka et al. (2022) | User Study | Users | 34 | 8 Weeks | Activity Scheduling Details, A... | Chatbot Feedbacks | IRB | Shapiro–Wilk Test, Mann–Whitne... |
| Ludin et al. (2022) | User Study | Users | 127 | 2 Weeks | / | PHD-9, GAD-7, PANAS, Acceptabi... | Yes | / |
| Fitzpatrick et al. (2017) | RCT | US University Students | 70 | / | / | Clinical Outcomes In Routine E... | IRB | Cohen's, ANCOVA, ANOVA |
| Noble et al. (2022) | User Study | Snowball Sampling, Social Medi... | / | / | Effectiveness, Engagement | Stress Levels, Sleep Quality,... | Yes | / |
| Mauriello et al. (2021) | User Study | Word Of Mouth And a University... | 47 | 1 Week | / | USE Questionnaire, Perceived B... | IRB | Wilcoxon Signed-Rank Test |
| Chung et al. (2021) | User Study | From Clinic, Snowball Sampling | 15 | 1 Week | User's Utterances | User Ratings | IRB | Spearman Correlation, Shapiro-... |
| Morris et al. (2018) | User Study | User | 37169 | 2 Weeks | / | Content, Structure, And Design... | Yes | Chi-Square Analysis |
| Beilharz et al. (2021) | User Study | Social Media Outlets, Online P... | 17 | 2 Weeks | / | / | Yes | / |