# OpenReview forum: "An Integrative Survey on Mental Health Conversational Agents to Bridge Computer Science and Medical Perspectives"
_EMNLP/2023/Conference — EMNLP 2023 Main_

### Official Review · Reviewer_2fQL · 2023-08-11

**Soundness:** 5

**Excitement:**

5: Transformative: This paper is likely to change its subfield or computational linguistics broadly. It should be considered for a best paper award. This paper changes the current understanding of some phenomenon, shows a widely held practice to be erroneous in someway, enables a promising direction of research for a (broad or narrow) topic, or creates an exciting new technique.

**Paper Topic And Main Contributions:**

The paper is a literature review comparing research on mental health conversation agents in the computer science field and the medical field. This paper highlights the disparity between the two, especially including the differential focuses in the fields when researching conversation agents - CS focuses more on the technical aspects of the model while medical focuses more on the user. Moreover, the paper highlights the technological gap between the two fields, with CS using more advanced models compared to the medical field. This paper does well to discuss the gap between the fields which should collaborate on this matter

**Questions For The Authors:**

1) Were there any interdisciplinary work reviewed between CS and medical field?
2) Did you examine any psychology or psychiatric specific databases?

**Reasons To Accept:**

This paper is extremely well-written and highlights integral issues regarding research on mental health conversational agents. The authors objectively summarize and discuss how the two different fields are approaching the matter and make good recommendations to improve the matter. It is interesting and incredibly relevant in the modern day, especially with the rise of AI companion chatbots being distributed and used almost everywhere.

**Reasons To Reject:**

The authors could provide more detailed recommendations for the two fields moving forward, as well as some suggestions for interdisciplinary work on this matter. It may also have been beneficial to examine psychology or psychiatric-specific research databases (in addition to PubMed) considering they were looking at the use of CAs with mental health issues.

**Reproducibility:**

4: Could mostly reproduce the results, but there may be some variation because of sample variance or minor variations in their interpretation of the protocol or method.

**Reviewer Confidence:**

4: Quite sure. I tried to check the important points carefully. It's unlikely, though conceivable, that I missed something that should affect my ratings.

---

> ### Author Rebuttal · Authors · 2023-08-28
>
> Thank you very much for your review. We aimed to give a thorough summary and comparison between the CS and medical fields about the work on mental health conversational agents differently. By comparison, we were able to suggest possible enhancements to both communities and provide future directions. Since interdisciplinary works consists a smaller portion of our final list (20 over 102 based on author affiliations on papers; 7 from ACM, 2 from IEEE, and 11 from PubMed) and there is a large disconnect between CS and the medical field, we hope this paper can act as the bridge between two communities that help to learn from each other. We will add affiliations in the feature list on the camera-ready version.
>
> We examined psychology-specific databases, like APA PsycNet, for keyword searching. However, we found that the results were minimal (4 in total) and existed in PubMed search results. So we did not include other medical databases in our criteria.

---

### Official Review · Reviewer_5Bp6 · 2023-08-12

**Soundness:** 4

**Excitement:**

3: Ambivalent: It has merits (e.g., it reports state-of-the-art results, the idea is nice), but there are key weaknesses (e.g., it describes incremental work), and it can significantly benefit from another round of revision. However, I won't object to accepting it if my co-reviewers champion it.

**Paper Topic And Main Contributions:**

The paper conducts a systematic review of mental health conversational agent(CA) papers and discuss the gap between the usage of conversational agent in mental health and the advance of language model in general AI research. They authors curated paper from multiple databases and conducted a multi-step filtering procedure to collect literature that match the topic. Distribution of multiple features from the finalist including languages, model techniques and target users were shown in the paper. The authors conclude there is a disparity in research focus and evaluation measurement between mental health and CS CA research but also have some common weakness in transparency. The authors call for evolving more recent technologies for mental health applications to bridge the gap.

**Reasons To Accept:**

1. Thorough overview of different aspects of conversational agent research in mental health and computer science.
2. Identify the gap that mental health CA has not benefited from most recent development of NLP methodologies.

**Reasons To Reject:**

1. The inclusion criteria is not inclusive enough. Consider i) other proceedings such as COLING, CHI and INTERSPEECH that also includes research work about conversational agent. ii) Expand the keyword list to include 'dialogue system', which may increase the completeness of the paper list.

2. As pointed out in the paper by the authors,  the intersection of mental health and conversational agent is still limited. The technical part of the paper is mostly about RNN, Transformers or rule-based approaches, so I feel like this topic may not be of great interest to the general audience of EMNLP.

**Reproducibility:**

4: Could mostly reproduce the results, but there may be some variation because of sample variance or minor variations in their interpretation of the protocol or method.

**Reviewer Confidence:**

4: Quite sure. I tried to check the important points carefully. It's unlikely, though conceivable, that I missed something that should affect my ratings.

---

> ### Author Rebuttal · Authors · 2023-08-28
>
> Thank you very much for your thoughtful suggestions. It is an excellent suggestion for the study to be inclusive of proceedings such as COLING, CHI, and INTERSPEECH. COLING is already included in ACL Anthology (referred to as ACL in the paper) and CHI is included in ACM sections reviewed in our paper.  Following the reviewer’s suggestion, we searched for publications from INTERSPEECH in the DBLP and the ISCA Archive, but unfortunately there is no paper in INTERSPEECH that has (“chatbot” or “conversational agent” ) and (“mental health” or “depression”) at the same time. We have also created an additional table summarizing the paper counts from each of the venues indexed in the 5 databases we surveyed. ACL Anthology (referred to as ACL in the paper) indexes 63 venues (ACL, NAACL, EMNLP, COLING, etc.), ACM covers over 40 journals and 270 conferences (including CHI related venues), IEEE over 200 journals and 2700 conferences, AAAI over 26 conferences, and PubMed covers over 5600 journals. The table below shows venues in each database, and corresponding experiment papers in our final list.
>
>
>
> | AAAI  |       | ACL     |       | ACM             |       | IEEE   |       | PubMed               |       |
> |-------|-------|---------|-------|-----------------|-------|--------|-------|----------------------|-------|
> | Venue | Count | Venue   | Count | Venue           | Count | Venue  | Count | Venue                | Count |
> | HCOMP |     2 | EMNLP   |     1 | CHI             |     9 | ICIRCA |     2 | JMIR Form Res        |     9 |
> | AAAI  |     1 | SIGDIAL |     1 | ACM-TiiS        |     4 | ACII   |     2 | J Med Internet Res   |     7 |
> |       |       | BioNLP  |     1 | IVA             |     4 | Others |    11 | Front Digit Health   |     4 |
> |       |       | NAACL   |     1 | ACM-HCI         |     3 |        |       | JMIR Mhealth Uhealth |     3 |
> |       |       | NLP4PI  |     1 | UbiComp-ISWC    |     2 |        |       | JMIR Res Protoc      |     3 |
> |       |       | LREC    |     1 | CUI             |     2 |        |       | Digit Health         |     2 |
> |       |       |         |       | PervasiveHealth |     2 |        |       | JMIR Ment Health     |     2 |
> |       |       |         |       | Others          |     9 |        |       | JMIR Hum Factors     |     2 |
> |       |       |         |       |                 |       |        |       | Internet Interv      |     2 |
> |       |       |         |       |                 |       |        |       | Others               |     9 |
>
> 'Others' in the table refers venues that only have one paper. They are:
> * ACM: CHItaly, ACSW, H3, Asian CHI, DIS, CHIuXiD, ACM-HEALTH, IASA, ECCE
> * IEEE: ICoICT, UCET, ICCCI, ICHCI, ICACCS, ISCC, IEEE Trans. Emerg., SIEDS, IEEE Pervasive Comput., ICCAS, INCET
> * PubMed: Curr Psychol , Comput Math Methods Med, Inf Process Manag, Front Psychol, Trials, Front Psychiatry, Drug Alcohol Depend, Sensors (Basel), JMIR Med Inform
>
> We are very grateful for your recommendation about the keywords. We included "dialog system" in the initial set of keywords in our review, but found that it only includes a small number of non-duplicated publications (7 from ACM, 0 from PubMed, 0 from IEEE, 7 from ACL, 3 from AAAI) compared to our current set of keywords and many of them did not satisfy our criteria for paper selection. Consequently, we excluded this keyword from our search. We will highlight this in the revised manuscript, given an opportunity. Below is our search result using "dialog system"
>
> |                     | AAAI | ACL | ACM | IEEE | PubMed |
> |---------------------|------|-----|-----|------|--------|
> | Search Result       |   10 |  13 |  79 |    0 |      1 |
> | After Deduplication |    3 |   7 |   7 |    0 |      1 |
> | Full-Text Screening |    1 |   0 |   1 |    0 |      1 |
>
> It is true that many of the papers we reviewed use traditional approaches for conversational agents. However, we want to state that our search criteria are between January 2017 to December 2022, which is before ChatGPT became famous. It is also worth noting that Medical domain papers implement rule-based systems based on psychotherapy or behavior change which are easy to program for controlled text generation, which we found not to be the case for CS papers. Our paper acts as an integrative review of the pre-LLM era studies in conversational agents, and provides insights for future studies in both CS and medicine fields. We found that there were specific reasons why previous studies and applications hesitated to use generative models (See Section 5.2). For instance, uncontrolled responses from LLM could possibly raise safety concerns in addition to ethical concerns. Technical requirements also were reported to act as a barrier for the medical community. Addressing these issues could be one of the directions that the NLP community can take.

---

### Official Review · Reviewer_e2C5 · 2023-08-12

**Soundness:** 3

**Excitement:**

4: Strong: This paper deepens the understanding of some phenomenon or lowers the barriers to an existing research direction.

**Missing References:**

Authors are advised to dive a little deeper into the available study and opt for a holistic approach to self-assess the selection of venues for PRISMA.

**Paper Topic And Main Contributions:**

The paper is a comprehensive survey of 136/534 papers in the space of conversational agents pertaining to mental health (or virtual mental health assistants) published in both computer science and medical venues. In the attempt to summarize the paper's takeaways, the authors presented some interesting findings on transparency, ethics, and cultural heterogeneity, along with essential recommendations to help bridge the cross-domain gap.

**Questions For The Authors:**

How will the recommendations of this study be adapted to accommodate the current virtual mental health assistants, particularly in light of safety (safeAI), changing user preferences/behavior/symptoms, and the dynamic ethical considerations associated with mental health support?

**Reasons To Accept:**

1. The authors targeted an essential space to review with a very specific set of research questions to answer/understand (line 100 - 106).

2. Diligent selection of papers through the PRISMA framework.

**Reasons To Reject:**

While the paper displays a diligent collation and intelligently observed takeaways, here are a few concerns pertaining to this review.

1. Although the PRISMA framework filtered a significant number of papers from the 534 papers initially recorded. It is easy to see that authors clearly missed not just a few but a significant number of papers from other potential venues. Nonetheless, I would like to raise an important point regarding the potential inclusion of additional papers from alternative sources. Drawing from my experience in both medical and AI domains, I believe that the selection of papers from various venues (9 from ACL, 4 from AAAI, 20 from IEEE, 40 from ACM, and 63 from PubMed) might benefit from a more diverse and inclusive approach, thus reducing bias and including fairness in the study. The reasoning to this in line 665 does not realistically comply with the statement in line 663. With this, I am not concerned about the frameworks' selection but would like to know the author's views on opting for selective venues.

2. The findings are interesting; however, I find it inclined toward a limited selection of papers and hence carry a bias toward particular needs in the medical domain to improve. Emphasis should be on collaborative working in both fields (AI and Medical) and how it can be further nurtured. However, I would like to highlight the potential inclination of the paper toward addressing specific needs within the CA domain, leading to bias. To enhance the overall impact and applicability of the findings, I recommend placing greater emphasis on the importance of collaborative efforts between the AI and Medical fields, not just in CAs but in peer and multi-party conversational analysis as well. It will be a more holistic approach to addressing the challenges and opportunities identified in the paper.

3. While the paper attempts to bridge the gap between computer science and medical studies, specifically focusing on CAs, the pool selection raises concerns about the generalizability of the findings. As it stands, the paper's conclusions and recommendations could be seen as disproportionately influenced by the chosen subset, diminishing their applicability and relevance to the wider research community.

**Reproducibility:**

N/A: Doesn't apply, since the paper does not include empirical results.

**Reviewer Confidence:**

5: Positive that my evaluation is correct. I read the paper very carefully and I am very familiar with related work.

**Typos Grammar Style And Presentation Improvements:**

-

---

> ### Author Rebuttal · Authors · 2023-08-28
>
> Thank you for your review and suggestions. It seems that you have specific venues in mind that we are missing, can you please give us specific examples that we are missing which would cause bias?
>
> The scope of our paper is to review studies focused on conversational agents published across the CS and Medical fields and we believe the current databases (​​ACM, IEEE, ACL, AAAI, PubMed) cover most of the edge-cutting studies, if not all, in both CS and Medical fields. ACL Anthology (*CL) indexes over 42 ACL and 21 Non-ACL conferences, journals, and workshop proceedings in the computational linguistics and natural language processing domain (ACL, EMNLP, NAACL, EACL, COLING, among others), ACM Digital Library publishes proceedings of over 40 journals, 270 conferences, and 9 magazines spanning several CS domains including HCI and computational linguistics research, IEEE indexes over 200 journals and 1700 conference proceedings specializing in engineering disciplines, AAAI covers over 26 proceedings such as ICML, AAAI, IAAI among others in the AI field, and PubMed includes journal articles published in 5600 journals with a specific emphasis on biomedical studies. We believe our selection criteria cover most papers dealing with conversational agents for mental health. We examined psychology-specific databases, like APA PsycNet, for keyword searching. However, we found that the results were minimal (4 in total) and existed in PubMed search results. We request the reviewer to highlight the specific venues that we missed leading to a bias in the study, and we are excited to add them to our list. We followed and expanded on the venue selection of the CS field from a previous survey paper ([Valizadeh and Parde, 2022](https://aclanthology.org/2022.acl-long.458.pdf)), which also has a similar range of coverage in the final list. However, our paper differs in the following ways: 1. We encompass all categories of conversational agents, not just those that are task-oriented, 2. We incorporated PubMed in our database selection to provide an integrative view of two domains, 3. Our focus is on mental health conversational agents, not all healthcare applications.
>
> Ethical considerations are one of the most important focus in our feature selection. Among our full list of 102 experiment papers, 70 papers mention ethical considerations, and 26 papers are IRB-approved. While data privacy and mentions of violence in language are the main concerns among researchers, we found that previous studies, especially in the medical community, choose not to use large language models because of wanting complete control over dialogue generation. From our study of over 100 papers, we belive addressing challenges such as biases, non-deterministic nature of generative models, and input/output data security should be the main focus for the NLP community to help LLMs to be broadly accepted in real-world situations.
>
> For further explanation of line 663 and 665, we found that CS databases had a large number of papers on the database search stage, but passed a smaller number of papers at last, which is significantly different compared to the inclusion ratio in PubMed (17.0% vs 60.6%). This is due to many papers from CS databases mentioning mental health as an important application while introducing background and application of conversational agents, but did not design a mental health-specific solution. This phenomenon shows that the CS community understands the importance of conversational agents as a support to help with mental health issues, but did not give an equivalent amount of effort in model design. Our paper aims to reveal this gap and raise attention to CS communities.
>
> Although changing the paper's main focus from conversational agents to general AI could include more papers in the final list, that diverges from the focus of this paper on conversational agents, given the rise of focus and technology in large language model driven chatbots. Furthermore, our paper aims to provide key takeaways to the *CL community by summarizing and comparing how two different fields, CS and Medicine, approach this task differently.

---

### Meta-Review · Area_Chair_WN9T · 2023-09-14

**Recommendation:** 4

**Metareview:**

The paper provides a survey on mental health chatbots, paying particular attention to "bridging the gap" between medical and computer science perspectives.

**Pros**: Reviewers all agree on the importance of the problem. Most reviewers highlight the paper as well-written with clear takeaways, bringing light to an understudied area of health-oriented conversational agents (i.e., for mental health). Reviewers find the survey to be "thorough" and "diligent".

**Cons**: Regarding the last "pro", two reviewers agree the survey is comprehensive, but express concerns about potential selection bias (w.r.t the papers chosen) due to an apparently limited number of papers from important venues. Authors provide a detailed rebuttal with exact counts from mentioned venues, but some reviewers still express minor concerns about this. One reviewer would have liked to see even more venues, while another reviewer was still struggling to make sense of the numbers (based on their existing experience). In any case, these concerns may be more minor as evidenced by the scores.

---

### Decision · Program_Chairs · 2023-10-07

**Decision:**

Accept-Main

**Comment:**

The paper provides a survey on mental health chatbots, paying particular attention to "bridging the gap" between medical and computer science perspectives.

**Pros**: Reviewers all agree on the importance of the problem. Most reviewers highlight the paper as well-written with clear takeaways, bringing light to an understudied area of health-oriented conversational agents (i.e., for mental health). Reviewers find the survey to be "thorough" and "diligent".

**Cons**: Regarding the last "pro", two reviewers agree the survey is comprehensive, but express concerns about potential selection bias (w.r.t the papers chosen) due to an apparently limited number of papers from important venues. Authors provide a detailed rebuttal with exact counts from mentioned venues, but some reviewers still express minor concerns about this. One reviewer would have liked to see even more venues, while another reviewer was still struggling to make sense of the numbers (based on their existing experience). In any case, these concerns may be more minor as evidenced by the scores.